# A Review of Virtual Reality for Individuals with Hearing Impairments

Stefania Serafin [1,*] , Ali Adjorlu [1] and Lone Marianne Percy-Smith [2]

1 Multisensory Experience Lab, Aalborg University Copenhagen, 2450 Copenhagen, Denmark
2 Copenhagen Hearing and Balance Center, Ear, Nose and Throat (ENT) and Audiology Clinic, Rigshospitalet, Copenhagen University Hospital, DK-2100 Copenhagen, Denmark
* Correspondence: sts@create.aau.dk

**Abstract:** Virtual Reality (VR) technologies have the potential to be applied in a clinical context to improve training and rehabilitation for individuals with hearing impairment. The introduction of such technologies in clinical audiology is in its infancy and requires devices that can be taken out of laboratory settings as well as a solid collaboration between researchers and clinicians. In this paper, we discuss the state of the art of VR in audiology with applications to measurement and monitoring of hearing loss, rehabilitation, and training, as well as the development of assistive technologies. We review papers that utilize VR delivered through a head-mounted display (HMD) and used individuals with hearing impairment as test subjects, or presented solutions targeted at individuals with hearing impairments, discussing their goals and results, and analyzing how VR can be a useful tool in hearing research. The review shows the potential of VR in testing and training individuals with hearing impairment, as well as the need for more research and applications in this domain.

**Keywords:** virtual reality (VR); hearing impairment; hearing aid; cochlear implant; training

## 1. Introduction

We live in a multisensorial world. Research shows how the human senses do not act as isolated modalities, but they influence each others [1–3]. Classical examples of interactions among the senses are the ventriloquism effect [4] where spatially conflicting audio-visual objects are judged to be placed at the location of the visual counterpart, and the McGurk effect where what one person hears is determined by the lips' movement of what a person says [5]. Such examples show the dominance of vision over audition. The sense of hearing is highly affected not only by vision but also by other senses such as touch and proprioception [6,7]. As an example, it has been shown that tactile information in the form of a puff of air facilitates speech intelligibility [8].

Traditional testing methodology performed in a sound isolated booth and stimulating only the auditory modality lacks the so-called ecological validity. For example, they cannot be translated to the real world where subjects are exposed to multisensory stimuli that interact together [9]. As an additional challenge, when examining hearing aids it has been observed that there exists a disparity between their performance in laboratory settings versus the real world [10]. As a matter of fact, some individuals perform well when their hearing is tested in laboratory settings, but they have difficulties following conversations in everyday life, where the sonic environment is complex.

These observations demand novel ways to test hearing capabilities that go beyond traditional testing methodologies.

Virtual reality (VR) technologies can provide a solution to ecological validity. Such technologies are becoming affordable and user-friendly, both from the hardware and software perspective, in such a way that they can be taken out of laboratories and used

in clinical settings. VR allows us to create simulations that imitate the real world while keeping a degree of control that is not possible in the real world. Studies performed in VR can also be easily reproduced, as opposed to those running in the real world that is unpredictable. This element can potentially allow us to run studies in different laboratories and compare the results. Moreover, VR environments are easily modifiable, so it is possible to investigate the effects of modifying different elements such as increasing noise conditions in intelligibility studies, or change the reverberation properties of a space, or any other variables [11]. In clinical settings, VR has been successfully used for different applications such as pain distraction [12,13], advance assessment and prevention of PTSD [14,15] and balance training [16], among others.

In [17], Patou questions whether clinical audiology is ready for VR, providing a positive answer. He observes the lack of ecological validity in clinical interventions, including rehabilitation. As an example, rehabilitation strategies are still too often designed to optimize performance on unrealistic assessment methods. This eventually proves ineffectual for transferring the benefit to real-world listening environments, to the detriment of users worldwide. He claims that VR is the ecological validity medium of excellence, combining a high level of control with a high degree of realism.

Globally, over 1.5 billion people are currently experiencing some degree of hearing loss (HL). WHO predicts that by 2050 this number will rise to 2.5 billion [18]. HL is classified into various stages based on degrees, ranging from mild, moderate, severe, or profound. Affected individuals are considered hard of hearing. (HoH) if they fit into any of the aforementioned stages [18]. There are multiple ways to combat HL. Assistive hearing technologies can enhance sound perception for hard of hearing (HoH) individuals via hearing aids or cochlear implants [19]. There are demonstrated practices to train sound perception with audiologic rehabilitation for HoH [20]. Both assistive hearing technologies and audiologic rehabilitation practices are primarily designed to facilitate speech and language perception, most vital for involving HoH in everyday communication. Some new solutions based on VR are appearing, also for music training. Such solutions will be discussed in Section 2.2.

In this paper, we first provide a review of approaches considering VR for hearing research. We then present some considerations on the use of VR in clinical settings.

## 2. Previous Work

In this section, we review previous and current research where VR is used to test or train the hearing skills of HoH individuals. The selected papers fit the following criteria:

- Methods include VR-based approaches with materials presented via a head-mounted device (HMD).
- The papers either use individuals with hearing impairment as test subjects, or present VR solutions targeted to individuals with hearing impairment.

This second point is particularly important, since we identified several papers targeted to hearing-impaired individuals, but that tested normal hearing individuals. We decided not to include such papers in the review, since we were specifically interested in papers addressing hearing-impaired individuals as test subjects.

The literature was identified through electronic searches by conducting extensive searches in the databases Web of Science: Science Citation-Index Collection (Thomson Reuters), Elsevier Scopus, ACM Digital Library, IEEE Xplore, and Google Scholar. The search was performed on September 2022 as per the logical expression: Title/ Keywords/Abstract contains ("virtual reality" "VR" OR "virtual environments" OR "VR" OR "Virtual Experience") AND ("audiology" OR "hearing research").

After screening the papers according to the criteria presented above, 18 papers were found. Such papers cover the topics of measuring and training hearing skills as well as gamified training and increasing accessibility for individuals with hearing impairment using VR. The papers are classified in Table 1, according to the kind of sound delivery method used (loudspeakers versus headphones), visual feedback used (360 degrees cap-

tured footage or 3D rendered), if the VR experience was used inside or outside laboratory settings, and if the application developed was for testing, training or increasing accessibility. Overall, we noticed a preference for the use of 3D generated content versus 360 degrees (12/18), and a less frequent use of loudspeakers versus headphones (6 studies using loudspeakers, 11 studies using headphones, and 1 study using both), and a more frequent use of laboratory testings rather than testing in the real world (10/18). In total, 7 out of 17 studies involved testing hearing capabilities, 7 studies involved training, and 4 studies presented accessible solutions for individuals with hearing impairment using VR. It is again important to notice that the literature presents additional papers where VR is used for hearing research, but these papers were not considered since they did not test hearing impaired individuals.

**Table 1.** Summary of the projects analyzed in this paper, including the sound delivery method (loudspeakers versus headphones), the visual content (3D versus 360 degrees video) if the application works only in laboratory settings or also in a real-life situation, and if it is used as a measuring or training tool.

| Reference | Reproduction Device | Visual Rendering | Test Scenario | Study Purpose |
|---|---|---|---|---|
| [21] | Loudspeakers | 360 | Lab | Testing |
| [22] | Headphones | 360 | Lab | Testing |
| [23] | Headphones | 3D | Lab | Testing |
| [24] | Loudspeakers | 3D | Lab | Training |
| [25] | Loudspeakers | 360 | Lab | Testing |
| [26] | Loudspeakers and headphones | 3D | Lab | Testing |
| [27] | Headphones | 3D | Real life | Training |
| [28] | Headphones | 3D | Lab | Training |
| [29] | Headphones | 3D | Real life | Training |
| [30] | Loudspeakers | 3D | Lab | Testing |
| [31] | Headphones | 3D | Real life | Training |
| [32] | Headphones | 3D | Real life | Training |
| [33] | Headphones | 3D | Lab | Testing |
| [34] | Loudspeakers | 33 | Lab | Training |
| [35] | Headphones | 360 | Real life | Accessibility |
| [36] | Headphones | 3D | Real life | Accessibility |
| [37] | Headphones | 360 | Real life | Accessibility |
| [38] | Headphones | 3D | Real life | Accessibility |

## 2.1. Testing Spatial-Hearing Skills

Ref. [21] presents a VR-based audiovisual paradigm with augmented real-life recordings, where HoH individuals are asked to perform a speech-in-noise test. The goal of the experiment is to ask participants to evaluate a set of hearing aid programs. The environment contains four competing talkers and was recorded with a 360-degree video camera and an Ambisonic microphone. The video was displayed in an Oculus Go VR headset, while the audio was presented to the participants via a circular loudspeaker array. Additionally, a fixed avatar was augmented in the scene, in order to use it as a placeholder for the delivered speech. The simulator sickness questionnaire [39] was used to evaluate the sickness symptoms of HoH individuals experiencing the setup. Furthermore, a questionnaire was used to assess the reproduction quality and outcome expectation. A total of 27 HoH adults were tested. Results showed that participants appreciated the realism of the VR intervention. Additionally, the participants reported a low degree of nausea and disorientation. The results of this investigation show promising possibilities regarding the use of VR for testing HoH individuals.

In Ref. [22], Udesen presents a VR platform developed at GN Resound to test directionality in hearing aids and headsets. The setup uses a virtual speaker array and a virtual

hearing device and video is captured using a 360 degrees camera. The whole setup runs in the platform Unity3D (www.unity.com, accessed on 22 November 2022). The auditory feedback can be delivered through headphones or speakers array. Specifically, the virtual hearing device and the virtual speaker array allow us to create fast prototypes of virtual spaces without having to deal with an expensive speaker array and a physical real-time prototype hearing device. The goal is to design a flexible and agile environment for laboratory testing of hearing aids as well as general spatial abilities using VR. This setup is now used and extended as part of the VR experimental lab of the GN Resound company in their Danish facilities.

In Ref. [23] the VR laboratory at Oldenburg University is presented. The aim of this laboratory is to create interactive and reproducible testing of subjects with and without hearing devices in challenging communication conditions. More specifically, until now in the laboratory five virtual environments have been designed: a cafeteria, a lecture hall, a train station, a street with car traffic, and a living room. These are typical environments where HoH individuals have generally expressed discomfort and challenges in listening to conversations. In these environments, virtual characters as well as virtual audio sources can be added. These environments are merely designed to perform speech recognition tasks, detection tasks, and divided attention tasks. As in other research projects, VR allows flexibility as well as a degree of control. One of the challenges when designing complex VR environments is the design of realistic and compelling avatars to make the experience plausible for HoH individuals.

An extended description of a binaural auralization system to test individuals with hearing loss is presented in Ref. [24]. The idea is to be able to reproduce complex acoustic scenes both indoors and outdoors. This system can also generate complex virtual acoustic environments reproduced through loudspeakers. In addition, the system includes room acoustic simulations. Moreover, the system is also able to reproduce simulated hearing aids signals. To create a realistic simulation, the auralized scene is updated in real-time according to the user's movements that are tracked via a motion capture system. The paper carefully describes the different elements of the system, with attention to precision and accuracy as well as computational efficiency and minimal latency. At the time of writing, listening tests still need to be performed in order to evaluate the tradeoff between computational efficiency and accuracy.

In Ref. [40] a study was conducted to explore the efficacy of using VR technology in hearing research with children by comparing speech perception abilities in a typical laboratory environment and a simulated VR classroom environment. The study included 48 participants (40 children and eight young adults). The study design utilized a speech perception task in conjunction with a localization test performed in auditory-only and auditory–visual conditions. The visual environment was delivered through an Oculus Rift head-mounted display. Results show that as expected the speech perception scores were higher for the audio-visual conditions over the audio-only conditions across age groups. In addition, children's performance on the speech perception task in the VR classroom was more similar to their performance in the laboratory environment for audio-visual tasks than it was for audio-only tasks. These results suggest that VR head-mounted displays are a viable research tool in audio-visual tasks for children, increasing flexibility for audiovisual testing in a typical laboratory environment.

In Ref. [25], the feasibility of using a virtual space as a speech test instrument is investigated. As it was the case in Ref. [40], the hypothesis is that the ability of individuals to recognize speech improves when visual cues are provided. 30 individuals with normal hearing and 25 individuals with hearing loss completed a classical audiometric test using pure tone, as well as the Korean version of the Hearing in Noise Test. The participants listened to a target speech and repeated it back to the tester for all conditions. HoH individuals completed the test with and without visual feedback. Results showed that augmenting the speech with visual information had a significant impact on speech performance between normal hearing and HoH individuals. HoH individuals had a better

integration of audio and visual cues. Overall the test was perceived as positive, besides the aspect of the ergonomics of the headset, which was reported as being too heavy.

As previously mentioned, one of the advantages of VR is that the different environments can be tested in several locations, which increases sharing of research data and experiments. However, few VR experiences are freely available to experiment with. One exception is the work presented in Ref. [26], where an extendable set of complex auditory-visual scenes for hearing research that allows for ecologically valid testing in realistic scenes while also supporting reproducibility and comparability of scientific results is presented. Three virtual environments are provided (underground station, pub, living room), consisting of a detailed visual model, an acoustic geometry model with acoustic surface properties as well as a set of acoustic measurements in the respective real-world environments. The current data set enables audio-visual research in a reproducible set of environments. All the environments are freely available in zenodo (https://zenodo.org/communities/audiovisual_scenes/, accessed on 1 November 2022) and the site is open to future contributions. The authors also provide a text file for each environment, which describes the different parameters.

### 2.2. Training Hearing Skills

It is acknowledged in the literature that auditory training improves the performance of children with a cochlear implant [41,42]. In Ref. [27], a VR-based application to train musical skills is presented. The goal is to provide a way to feel and see music to people with hearing disabilities through a VR application, transforming the audio signals captured in a song into vibrations sent at a certain intensity through vibration motors. Three scenarios are evaluated: the first scenario where the user experiences the music through VR, a second scenario, where the user experiences the music through the vibrations, and the third scenario, where VR and vibrations are combined.

A preliminary usability evaluation shows the potential of training HoH individuals with a combination of visual and haptic signals, with the third scenario scoring significantly higher scores in usability and user experience when listening to music. In Ref. [34] a VR setup is used to test and train spatial hearing skills. Specifically, it is investigated if training can improve azimuth localization for bilateral cochlear implant users. With 20 users, the effects of two training procedures (spatial versus nonspatial control training) were assessed on two different tasks performed before and after training. Such tasks were head pointing to sound and audiovisual attention orienting. Spontaneous head movements while listening to the sounds were allowed and tracked to correlate them with localization performance.

During spatial training, the users reduced their sound localization errors in azimuth.

#### Gamified Training

An additional challenge when testing or training HoH individuals, especially in children, is that many tests and training programs are rather repetitive and boring, so the results of the tests and training might be affected by fatigue. One approach to cope with this issue is to adopt gamification as an extra element. To the best of our knowledge, Ref. [28] is one of the first studies that used gamification in VR to help HoH children. In Ref. [28], a 3D VR game was compared to a 2D game in training HoH children's spatial rotation skills. 44 HoH children (aged 8 to 11) participated in the experiment. The experiment was run using Virtual Boy by Nintendo, a video game console released in 1995. The results indicated that the 3D VR intervention had improved the spatial rotation skills significantly more than its 2D equivalent.

Research on training musical skills for HoH individuals using VR is still in its infancy. To our knowledge, Ref. [30] presents the first study of pitch perception for HoH children using VR. This study aims to develop a VR tool to compare the pitch ranking abilities between children with a cochlear implant, hearing aids, or normal hearing and to discuss the potential benefits of using VR in a clinical test setting. Furthermore, the study explored

if pitch ranking performance was affected by clinical or musical background factors. The results indicate that VR is effective in assessing the pitch ranking abilities of all participants, which included children with a cochlear implant, hearing aids as well as normal hearing. Additionally, the children enjoyed the excitement of trying VR technologies. Unfortunately, this excitement died out pretty quickly since children soon found the tasks boring and too long.

In September 2021, a two-year partnership project *Listen Again* started between the Center for Hearing and Balance (CHBC) at Rigshospitalet, the Multisensory Laboratory (ME-Lab) at Aalborg University (AAU) and Decibel—a patient organization for children and adolescents with hearing loss. The goal of the project is to use VR technologies and gamification to train HoH individuals in spatial awareness in everyday life. Two VR scenarios were created for this purpose: a school playground and a music museum. In the app, there are aspects of gamification, which optimize the children's motivation to train. The children are given VR glasses and train at home for 15 min twice a week for three months. Results show that children experience VR combined with gamification as a valuable technology to train their spatial awareness. However, gamified experiences need to become more engaging in order to extend the engagement of the children during the training period [31].

The BEARS (Both EARS) project has a similar goal to the Listen again project, specifically to develop a package of VR games to train spatial hearing in young people (8 to 16 years) with bilateral cochlear implants using an action-research protocol [29]. In BEARS a package of VR games has been developed to train teenagers with bilateral cochlear implants in sound localization and spatial-listening skills. The aim is to confirm whether the use of BEARS leads to improvements in everyday hearing. At the time of writing the results of the projects are not available yet. However, according to the plan, the project plans to recruit 384 children (8–16 years) with bilateral implants from 9 clinics. The children will be randomly allocated to one of two groups: BEARS or traditional care. The BEARS group will receive 3 months of spatial-listening training. Both groups will attend clinics for assessments at baseline, at 3 months, and at 12 months. Qualitative interviews will occur following the trial. Outcomes include spatial speech-in-noise measures, quality of life, resource use, and perceived benefits.

The purpose of the study presented in Ref. [32] was to investigate the effects of VR-based cognitive training in HoH older adults. The participants were three HoH older adults. Three assessment tools: audiometric, neuropsychological, and outcome measurements of the subjective hearing were used before and after the VR cognitive training. The VR cognitive training was conducted once per week for 6 weeks and consisted of five different VR games classified into three specific cognitive domains (attention, memory, and executive function). The experiment demonstrates that VR cognitive training could improve cognitive function, speech-in-noise perception, and subjective hearing in the hearing-impaired elderly. Being the content of the paper is in Korean, we translated using google translate. The three subjects performed VR-based cognitive training conducted once a week for a total of 6 weeks (60 min per session) at the Dongmyung Auditory Clinical Center at Dongmyung University. The training consisted of a total of 60 min per session, including a 5-min break for eye massage. Three training games were played twice each for the first 25 min, and two training games were played twice each for the next 20 min after an interim break. During the 6-week training period, all three subjects steadily improved their scores in the five cognitive training games.

### 2.3. VR for Accessibility

Another direction at the intersection of VR and HoH individuals is in the field of accessibility, specifically how to design VR environments that are usable.

In Ref. [33] EarVR is proposed as a way to support HoH individuals in their use of VR. The goal of EarVR is to create an assistive device that analyzes sounds in 3D and notifies the user about the direction the sound is coming from (azimuth angle). This is achieved by

using two vibro motors placed on the user's ears, notifying respectively of the left and right directions. EarVR was tested with HoH individuals. Results show that the solution helps HoH individuals to perform sound localization tasks in VR that they could not perform before, as well as increasing their encouragement to complete sound-related tasks in VR. In a related study, the same VR simulation is used as a novel method for the visualization of 3D spatial sounds [43,44].

Ref. [35] presents a study that evaluates the efficiency of a language processing system, which makes use of VR technology combined with AI implementations for automatic speech recognition (ASR), sentence prediction, and spelling correction. Selected participants were invited to a play, and after the performance, they answered four structured questions about image/display, subtitle, understanding, and satisfaction using a Likert-scale (1 poor to 5 best). Image/display (provided by a Samsung Gear VR) received the worst ratings, while subtitles were, overall, judged as positive. Overall the results show a positive attitude towards the system regarding understanding and satisfaction along the entire play sessions, but some improvements are wished towards the image quality and overall ergonomics of the setup.

It can be difficult for normal-hearing individuals to understand the issues encountered by individuals with hearing impairment. To cope with this issue, in Ref. [36] a 3D simulation of how it feels to be a child with a cochlear implant or a hearing aid in a school context is presented. In this simulation, the user can navigate in a school playground as well as sit in a classroom. The VR application was shown to parents of children with hearing impairment who found the tool useful since it allowed them to understand the challenges of their child during everyday activities such as going to school or a playground. Additionally, the parents indicated that this could be a useful tool for teachers, coaches, and professionals working with children with hearing impairment as they could get an insight into how their students perceive their lectures or sessions. Simulations do not completely represent the issues encountered by individuals with hearing impairment, since ultimately we hear with the brain and not the ears, and the brain adapts over time to the provided technology. However, simulations can at least provide an indication of the challenges present when a child is with peers in a school environment.

In Ref. [37] DAVEE is proposed as an accessible classroom environment for HoH individuals. The system is a VR simulation that addresses the challenges of HoH individuals through sign language and 360 degrees instructional videos. Accessibility in education for HoH individuals is also addressed in Ref. [38]. Here, a 3D tool to teach mathematics to HoH individuals is presented. One of the systems presented in the paper is called, Smile, and is an immersive virtual learning environment for children from 5 to 10 years. This is achieved by having characters that communicate through spoken English as well as American Sign Language. An evaluation showed that children found the system usable and enjoyable.

## 3. Considerations on the Use of VR for Clinical Research

In the following section, we discuss some considerations for the use of VR in clinical settings. These considerations are based on the previous literature review as well as on personal experience in using VR for pediatric audiology.

### 3.1. Hardware Selection

VR devices have achieved a level of democratization such that they can be purchased for a relatively low cost. Moreover, at the time of writing, devices such as the Oculus Quest 2 by Meta do not require additional hardware or a separate desktop computer, and can be given to a general audience to train at home. This has already been done in the Listen again project, where children received an Oculus Quest 2 HMD. The recently announced Oculus Quest Pro has also embedded eye tracking, which enhances the possibility of hearing research. As an example, pupillometry is widely used to quantify listening effort [45].

### 3.2. Auditory Displays

Regarding auditory displays, it is well known that high quality audio significantly contributes to an immersive experience [46]. The headphones embedded in commercially available HMDs are still rather low quality, and usually an external additional pair of headphones is used for higher fidelity auditory display.

From the software perspective, in recent years virtual acoustics has made tremendous progress, and VR applications include rendering of realistic auditory stimuli, including spatial rendering and room acoustics. However, challenges still exist in the simulation of efficient yet accurate auralizations. Main issues related to the fact that accurate room simulations are extremely computationally expensive, preventing a real-time implementation. Therefore techniques to create efficient yet accurate virtual auditory environments are an active area of research [47,48].

Moreover, the consideration that each user's individual head and pinnae shape affects the soundwave transmitted to the eardrum, or the so-called personalized head-related transfer function (HRTF), can be integrated into VR simulations [49,50]. The question of the utility of personalized HRTFs is, however, still open. For example, Ref. [51] shows how in an interactive VR simulation where several cues are present, such as visual, motion, and proprioception cues, generalized HRTF is sufficient. This can be due to the fact that localization and externalization cues are facilitated by the other sensory modalities.

Research also shows that the use of non-individualized HRTFs when using virtualization techniques does not impair perceived auditory distance [52].

Research on interactive auralization for VR is very active. The main goal is to provide simulations that are accurate yet efficient. Understanding human perception when exposed to audio-visual cues of different qualities is also an active topic of research [53].

### 3.3. Visual Displays

By replacing real-world visual information with digitally generated ones, VR-HMDs can place users inside relevant virtual environments where they can receive appropriate treatment for various mental- and physical disabilities [54]. This advantage of virtual displays can also be translated into hearing impairment research. Although several tools nowadays exist to generate content for high-fidelity visual displays [55,56], creating high-quality 3D graphics is still very time-consuming. One alternative solution is using a 360 degrees camera that can capture real scenes to be reproduced in an HMD [57]. The advantage of using recorded footage is obviously the fact that the content is quickly and readily available and a faithful reproduction of reality. One disadvantage is the fact that it cannot be modified, and the point of view depends on the location of the camera. On the other hand, 3D-generated content is highly adaptable and modifiable but more time-consuming to create and requires skills in computer graphics in order to generate realistic simulations.

### 3.4. Natural Interactions

One key attribute of VR is its interactivity, allowing the user to take a very active role. The VR community has for several decades extensively researched interaction techniques in 3D user interfaces [58]. Despite that, commercial VR devices are still based on joysticks that do not faithfully represent the complex interactions that humans can have in the real world. Some devices such as the Oculus Quest 2 have the possibility to replace joysticks with hand tracking, which allows us to increase the naturalness of the interaction. However, interacting with one's own hands removes both tactile and haptic feedback present in real-life interactions. In the physical world if we reach towards an object with our hands we can feel the texture, shape, temperature, and other physical properties [59]. This does not happen when interacting in VR. In Refs. [59,60], Lederman and Klatzy present several exploratory procedures that humans use to recognize the haptic properties of objects. These are typical movements that we perform to recognize several properties, such as substance-related properties such as texture, hardness, temperature, weight, and structure-related

properties such as weight, volume, the global and exact shape and functional properties such as part motion and specific function. Researchers propose that such exploratory procedures should be implemented also in designing haptic interfaces, to afford more natural interactions between humans and technology.

The studies examined the interactions that involve the hands. In our literature search we found only one study that included a dual task (listening while walking) performed with the feet [61]. The objective of Ref. [61] was to investigate the effect of age-related hearing loss on word recognition accuracy in a dual-task experiment. Specifically, this research tried to cope with the limitations of highly controlled hearing tests that do not show ecological validity. This goal is achieved by creating a test scenario where HoH individuals can walk and perform a dual task.

This is certainly a very promising area for future research since it is still a challenge to understand how individuals with hearing impairment behave while performing other actions, such as walking. We did not include this study in Table 1 since it uses a screen and not an HMD [61]. Overall safety measures would need to be taken to investigate how HoH individuals behave when walking in a VR, given the likely issues of stability and balance.

### 3.5. User Centered Design

Projects like Refs. [29,31] involve the users in the whole process from the design of the application to the iterative testing. As previously observed in Refs. [62–64], involving the relevant stakeholders and accounting for what they find important improves and targets the final application. To our knowledge, the first example of participatory design in the field of audiology is presented in Ref. [63]. Here a music training app is developed through a series of workshops run by the end users. In the first workshop some mockups were created in collaboration with the end users. This participatory design approach proved to be essential in order to create relevant solutions for the end user. From an audio engineer's point of view, it is not trivial to adopt a participatory design point of view, especially because several stakeholders are involved in the process, from the clinicians to the patients and their caregivers.

### 3.6. Gamification

Especially when children are involved, a gamified approach such as the one used in Refs. [29–31] helps motivating training. This requires the involvement of experienced game designers. In Ref. [30] it is observed how recruitment of participants was quick and efficient, since many were intrigued by the possibility of trying a VR setup. However, many found the task boring and too long and recommended feedback after each trial to confirm that they had understood the task and to keep up motivation. One reason might be that the experience was designed by engineers without a participatory design approach involving the target group.

### 3.7. Cybersickness

It is not uncommon for individuals not familiar with VR to experience cybersickness. Cybersickness symptoms cause severe discomfort and hinder the immersive VR experience. Sensory conflict theory explains that motion sickness in VR can be due to the mismatch between visual and vestibular senses [65]. Additionally, individuals with hearing loss often suffer from vestibular sensory problems [66], making this target group more prone to motion sickness. A common methodology to assess hearing loss is by using the simulator sickness questionnaire [39]. In Ref. [67] cybersickness in 360-degree head-mounted display VR is investigated. In traditional 360-degree VR experiences, translational movement in the real world is not reflected in the virtual world, and therefore self-motion information is not corroborated by matching visual and vestibular cues, which may trigger symptoms of cybersickness. A new Artificial Intelligence software designed to supplement the 360-degree VR experience with artificial six-degrees-of-freedom motion was implemented, which showed to reduce cybersickness. A study conducted by Impellizzeri et al. investigated

whether cybersickness negatively affects the outcome of VR rehabilitation interventions for individuals with Parkinson's disease [68]. The results indicate that individuals with Parkinson's were not more prone to cybersickness than healthy controls. However, to our knowledge, no studies are investigating whether cybersickness can reduce the effectiveness of VR interventions for individuals with hearing disabilities.

### 3.8. Addressing Personal Needs

The articles reviewed in this paper address several aspects of training and accessibility using VR. One important element in these situations is the fact that individuals are very different from each other, both in terms of hearing disability and in terms of preferences. From the point of view of hearing disability, each training and rehabilitation program should be designed in order to target the specific needs of the user. Moreover, when looking at musical training, it is important to observe that individuals have different preferences regarding the music they like to be exposed to, and this can highly affect their engagement in adopting the training programs [69].

### 3.9. Measurement

The projects described above adopt different measurement approaches, from testing the ability of subjects to recognize speech, to performance, and behavior. Measuring behavior in VR is nowadays facilitated by the inclusion of different sensors in the HMD, such as eye tracking and motion tracking.

In Refs. [70,71] a way to measure user behavior in realistically simulated virtual environments is presented. While the solution is tested only with normal hearing individuals, the intent is to use it to monitor individuals with hearing impairment. There is evidence that head movement can affect the performance of hearing aids algorithms [72]. Head movement can be easily recorded using sensors embedded in the HMDs. Another interesting feature of novel HMDs is the fact that they include an eye tracker and pupillometry. Eye tracking and pupillometry complement other methods by providing objective physiological measures of online cognitive processing during listening. Eye tracking records the moment-to-moment direction of listeners' visual attention, which is closely time-locked to unfolding speech signals, and pupillometry measures the moment-to-moment size of listeners' pupils, which dilate in response to increased cognitive load [73]. Another objective measurement that is becoming adopted in laboratory settings is EEG [74]. In Ref. [75] EEG and VR are combined as a tool to study the neurophysiological mechanisms of everyday language comprehension in rich, ecologically valid settings, specifically a simulated restaurant in VR. Objective measurements become more challenging when the VR experiences need to be taken home for example for longitudinal training.

## 4. Conclusions

In this paper, we presented a review of the existing research that utilizes VR to test and/or train individuals with hearing disabilities. We presented the different applications of VR for hearing research, and provide some considerations on the technological and user-centered issues encountered when introducing VR to the clinic. In the last decade, advances in hardware and software technologies have increased the use of VR in both laboratory and everyday settings. However, the use of VR in audiology and with special applications for individuals with hearing impairment is still rather limited. Nonetheless, VR shows to have potential both in testing hearing skills and also in training individuals with hearing impairments. The democratization of VR makes it a suitable tool for hearing research, together with the possibility of increasing the ecological validity of the tests. The audio-visual quality, ergonomics, and software tools available for VR technologies will also increase its use in clinical practice. There are still challenges associated with the use of VR, especially with an elderly population, namely the fact that VR completely isolates individuals from the physical world, which can be daunting. In this case, other solutions

for example based on augmented reality, where the virtual world is superimposed on the virtual world, might be more suitable.

**Funding:** This research was funded by Nordforsk, Nordic Sound and Music Computing (number 86892) and Velux foundation.

**Acknowledgments:** The authors would like to thank the Velux foundation that supported the Listen Igen project described in this paper.

**Conflicts of Interest:** The authors declare no conflict of interest.

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
