# Peer review of "A Review of Virtual Reality for Individuals with Hearing Impairments"

_mti, doi:10.3390/mti7040036_

Round 1

Reviewer 1 Report

In this study, authors focused on Virtual Reality technologies to aid in hearing research. They focus on the studies that used head-mounted VR devices, included individuals with hearing impairment as test subjects, or presented VR solutions targeted to individuals with hearing impairment. The authors grouped the studies into testing, training, and accessibility. They also provided some consideration of VR technologies in clinical settings. 

Major comments:

- The authors performed a literature review in the hearing impairment area where VR is used via a head-mounted device in the study. They were able to find 18 papers to fit the criteria. This is a very narrow review and this shows that there is not enough study performed on the topic. I totally agree that there is no limit while putting together a review paper. However, the review requires depth in the topic. Are there other technologies similar to VR to combine?

- The authors mentioned the sources of their search. It would be better to include the search results. How many studies are found and how many of them are related to the topic? 

- The authors mentioned in Pg 2 lines 73-75 that " After screening the papers according to the criteria presented above, we collected 18 papers on the topics of measuring and training hearing skills as well as gamified training and increasing accessibility for individuals with hearing impairment using VR." Were there more than 18 papers and the authors selected 18 of them?

Minor comments:

- Section 0. Introduction Pg 1 line 54: "In this paper we first provide a review of the review of approaches considering VR for hearing research." 

-> Fix "In this paper, we first provide a review of the approaches considering VR for hearing research."

- Section 1. Previous work Pg 3 line 105-106: "The whole setup runs in the platform Unity3D [?]." -> missing the citation number

- Section 1. Previous work Pg 4 line 145-146: "The articipants listened to a target speech and repeated it back to the tester for all conditions." -> fix the second word to "participant"

- Section 1. Previous work Pg 5 line 194-195: "... - a patient organosation for children and adolescents with hearing loss." -> fix the second word to "organization"

Author Response

In this study, authors focused on Virtual Reality technologies to aid in hearing research. They focus on the studies that used head-mounted VR devices, included individuals with hearing impairment as test subjects, or presented VR solutions targeted to individuals with hearing impairment. The authors grouped
the studies into testing, training, and accessibility. They also provided some consideration of VR technologies in clinical settings.
Major comments

  • - The authors performed a literature review in the hearing impairment area where VR is used via a head-mounted device in the study. They were able to find 18 papers to fit the criteria. This is a very narrow review and this shows that there is not enough study performed on the topic. I totally agree that there is no limit while putting together a review paper. However, the review requires depth in the topic. Are there other technologies similar to VR to combine?

Answer: We prefer to consider VR only and not include other technologies.

- The authors mentioned the sources of their search. It would be better to include the search results. How many studies are found and how many of them are related to the topic?

Answer: We found precisely 18 studies which are the ones reported.

>- The authors mentioned in Pg 2 lines 73-75 that " After screening the papers according to the criteria presented above, we collected 18 papers on the topics of measuring and training hearing skills as well as gamified training and increasing accessibility for individuals with hearing impairment using VR." Were
there >more than 18 papers and the authors selected 18 of them?

Answer: No we found only 18 papers.

Minor comments:
Section 0. Introduction Pg 1 line 54: "In this paper we first provide a review of the review of approaches considering VR for hearing research."

Answer: Fixed

Fix "In this paper, we first provide a review of the approaches considering VR for hearing research."
Answer:  Fixed

Section 1. Previous work Pg 3 line 105-106: "The whole setup runs in the platform Unity3D [?]." ->
>missing the citation number

Answer: Fixed. It was supposed to be a footnote not a citation.

>- Section 1. Previous work Pg 4 line 145-146: "The articipants listened to a target speech and repeated it
>back to the tester for all conditions." -> fix the second word to "participant"
Answer:  Fixed

- Section 1. Previous work Pg 5 line 194-195: "... - a patient organosation for children and adolescents with hearing loss." -> fix the second word to "organization"
Answer:  fixed

Reviewer 2 Report

Summary

A review of how VR is utilized in the context of hearing impairment research is presented. A range of aspects of the role of VR is explored, including research into spatial-hearing skills, and gamified training. Considerations of the use of VR in clinical settings, such as the personal needs of individuals with hearing loss, is also discussed.

General comments

The paper is timely and interesting, and will be of interest to readers of Multimodal Technologies and Interaction. However, in my view the manuscript needs to be improved, and more detail is needed is several places where the text is sometimes too short or vague. In the current version, for quite a few of the studies cited, there is not much description beyond something like “Study X describes a virtual environment which allows laboratory testing of hearing aids as well as general spatial abilities using VR,” or “The goal of this study is to…” but this is not sufficient. There needs to be a clear description of how these studies utilize their virtual environments in hearing research and what they found, in order to demonstrate empirically that they are effective in reproducing real world environments in order to test auditory abilities among hearing-impaired individuals. At one point (line 222), a study is described where the author(s) were unable to read beyond the abstract because the manuscript was written in Korean. In other sections no citations are present. This is not sufficient for a scientific article.

While I feel that these issues can addressed, some substantial changes need to be made to the manuscript. I have added some comments below, that I hope will be useful to the author(s). With some additional work, I feel that this will be a worthwhile paper that will contribute to the literature.

Specific comments

Title: “Virtual Reality in Hearing research: current research, possibilities and limitations”

            Upon reading the paper, it becomes clear that the focus is on hearing loss research specifically, rather than use of VR in “normally hearing” populations (clarified on lines 63-66). I would suggest changing the title to reflect this e.g. “Virtual Reality in hearing impairment research: Current research, possibilities and limitations.”

Abstract: Currently, the abstract is a bit vague when it comes to describing what the focus of the work was, and the main conclusions that can be drawn. This section might be strengthened by adding some concise text clarifying the inclusion criteria for the papers described in the review, in line with the text on line 59 that the papers had to include VR-based approaches with materials presented via a head-mounted device (HMD), and used individuals with hearing impairment as test subjects, or presented VR solutions targeted to individuals with hearing impairment.

            A sentence or two summarising the main conclusions drawn by the paper, and/or suggestions for further work based on the work described, would also be helpful here.

Keywords: These are missing. Please add them.

Line 20: “As an additional challenge, when examining hearing aids it has been observed that there exists a disparity between their performance in laboratory settings versus the real world [6].”

            More detail please. What is the nature of the disparity? Is performance in the real world significantly worse?

Line 30: “Moreover, VR environments are easily modifiable, so it is possible to investigate for example the effects of increasing noise conditions in intelligibility studies, or change the reverberation properties of a space, or any other variable.”

            The statements about intelligibility and reverberation need supporting citations e.g. “…change the reverberation properties of a space (Kolarik et al. 2013)…” Kolarik, A. J., Cirstea, S. & Pardhan, S. Discrimination of virtual auditory distance using level and direct-to-reverberant ratio cues. J. Acoust. Soc. Am. 134, 3395–3398 (2013).

Line 54: “In this paper we first provide a review of the review of approaches considering VR for hearing research.”

            Minor point - suggest “In this paper we first provide a review of the approaches considering VR for hearing research.”

Line 99: “Results showed that participants appreciated the realism of the VR intervention and generally accepted it as a valid speech-in-noise test. Additionally, the participants reported a low degree of nausea and disorientation. The results of this investigation show promising possibilities regarding the use of VR for testing HoH individuals.”

            This is a little vague, and could use more detail. What does it mean that “participants…generally accepted it as a valid speech-in-noise test”? Shouldn’t the validity of the speech-in-noise test be assessed and evidenced by the authors of that study, rather than via subjective judgments of participants? What “promising possibilities” specifically are being referred to here?

Line 106: “platform Unity3D [? ].”

            There is a missing citation here.

Line 109: “The goal is to design a flexible and agile environment for laboratory testing of hearing aids as well as general spatial abilities using VR.”

            Again, more detail please. In the author’s opinion, was this goal met? What evidence was presented to the paper to demonstrate that this goal was achieved? Were any experiments performed, and what were the results?

Line 120: “As in other research projects, VR allows flexibility as well as a degree of control. One of the challenges when designing complex VR environments is the design of realistic and compelling avatars to make the experience plausible for HoH individuals.”

            This is an example of very general text being presented, where more specific text describing the findings of the study being described would be more helpful. It is not enough to mention that in [15] “virtual characters as well as virtual audio sources can be added” (line 118). There needs to be evidence of some evaluation of the effectiveness of the virtual environment for experimental testing. Experiments and their results should be described to illustrate this.

Line 124: “An extended description of a binaural auralization system to test individuals with hearing loss is presented in [16]. This system can also generate complex virtual acoustic environments reproduced through a loudspeaker. In addition, the system includes room acoustic simulations.”

            This is too short, and the description is not sufficient. The reader will want to know what is meant by “complex” virtual environments, what sort of acoustic simulations are being referred to, and whether or not the system has been tested to assess if it does an effective job at simulating real world situations.

            In contrast, the following paragraph describing [31] is much more useful to readers in reporting use of VR technology for hearing impaired participants, and highlights its use in research with children.

Line 159: “iii) a condensation point for future extensions and contributions for developments towards standardized test cases for ecologically valid hearing research in complex scenes.”

            Please add some explanation regarding what a “condensation point” is. In the lines above, some mention regarding how the “extendable set of complex auditory-visual scenes for hearing research that allow for ecologically valid testing in realistic scenes while also supporting reproducibility and comparability of scientific results” (line 152) was assessed to ensure that it did indeed allow for ecologically valid testing of realistic scenes.

Line 167: “A preliminary usability evaluation shows the potential of train HoH individuals with a combination of visual and haptic signals.”

            Too vague. What were the results of the study? Did training result in a significant increase in the abilities tested?

Line 200: “At the time of writing, the results of this training are under investigation [23].”

            This comes across in the text as on-going but unfinished research. Looking at the reference list, there is a publication for [23], but if the results are still preliminary and have not been analysed then this study probably should not be included in the paper. If the text can be updated to show what the findings were and that the VR environments and training were effective, then it should be included. Also, it is unclear what the tasks were and what the children were being trained to do.

Line 222: “Unfortunately only the abstract of the paper is written in english, the rest of the paper is in korean, so it is hard to discuss the methodology and results.”

            If the author(s) were unable to read the manuscript, then I would suggest that they should not include it in their paper. One of the purposes of review sections in papers is that the author(s) themselves assess whether the study is sound and report accordingly. If it was not possible to read the paper, how can the author(s) be able to ascertain that the study was designed and carried out correctly, and that the analyses and interpretations of the findings were appropriate?

Line 231: “Results show that the solution helps HoH individuals to perform some tasks they could not perform before.”

            Vague. What tasks are being referred to?

Line 237: “A quantitative and qualitative study was performed and demonstrated good overall results of the system regarding understanding and satisfaction along the entire play sessions.”

            Insufficient detail. What tasks were performed in the study? What exactly were the results? How was understanding and satisfaction measured?

Line 244: “The VR application was shown to parents of children with hearing impairment, who found the tool useful.”

            More detail please, as this is the important sentence in this paragraph. What constitutes “useful?” What aspects did the parents find useful?

Line 278: “However, challenges still exist in the simulation of efficient yet accurate auralizations [38,39].”

            Please state what challenges are being referred to.

Line 293: “For example, [42] shows how in an interactive VR simulation where several cues are present, such as visual, motion and proprioception cues, generalized HRTF are sufficient. This can be due to the fact that localization and externalization cues are facilitated by the other sensory modalities.”

            This is fine, however the different aspects of localization (azimuth, distance, elevation) is not mentioned here. To strengthen this point, I would also suggest adding another cited study here showing that use of non-individualized HRTFs when using virtualization techniques does not impair perceived auditory distance:

Prud’homme L, Lavandier M (2020) Do we need two ears to perceive the distance of a virtual frontal sound source? J Acoust Soc Am 148:1614–1623

Line 290: “2.3. Visual displays”

            This section does not contain any citations, and these should be included to strengthen the points made. It also needs to be clarified here why visual displays play an important role in hearing impairment research.

Line 300: “2.4. Interaction”

            Interaction might refer to many things, but here it is used in the context of haptic displays. Perhaps the subheading title could be changed to reflect this.

Line 326: “We did not include this study in the survey since it uses a screen and not an HMD.”

            This is confusing. What study is being referred to?

Line 358: “2.8. Addressing personal needs”

            A supporting citation or two here would help strengthen the points being made, especially for statements such as “individuals have different preferences regarding the music they like to be exposed to, and this can highly affect their engagement in adopting the training programs.”

Line 390, Conclusions section: “We presented the different applications of VR for hearing research, and provide some considerations on the technological and user centered issues encountered when introducing VR to the clinic. As the paper shows, VR has proven to be valuable in different aspects of hearing research, from measuring to training.”

            In my view, this is too weak to constitute an appropriate conclusions section. A summary of main “take home messages” of the paper would help the reader here. For example, what can be concluded regarding the current status of use of VR in hearing impairment research? What are the most useful aspects of VR in this context, and what are the strengths of using VR as an approach to hearing research? What areas of future VR research need to be explored? Are there any weaknesses associated with using VR in studies that need to be addressed?

Author Response

We are very grateful to R2 for the detailed feedback on our paper.

All the useful feedback from R2 was addressed.

See attached.

Reviewer 3 Report

The authors wrote a literary review about Virtual Reality in Hearing research. The review is well written and well organized, but I think it should not give a real contribution to literature.

1. Please, use the PRISMA criteria to selected the manuscripts reviewed ( how MDPI suggest)

2. Regarding the chapter of Cybersickness, I think that it should be improved, because often hearing problems are linked to vestibular problem. Please use this reference : Impellizzeri F, Naro A, Basile G, Bramanti A, Gazia F, Galletti F, Militi D, Petralito F, Calabrò RS, Milardi D. Does cybersickness affect virtual reality training using the Computer Assisted Rehabilitation Environment (CAREN)? Preliminary results from a case-control study in Parkinson's disease. Physiother Theory Pract. 2022 Nov;38(13):2603-2611.

Author Response

Thank you for your useful comments. We added the suggested reference.

Using PRISMA would require us to completely restructure the paper so we decided to keep the selection criteria from the first revision.

Round 2

Reviewer 1 Report

The authors addressed all of the issues.

Author Response

Thank you for the positive feedback.

Reviewer 2 Report

I appreciate the efforts of the author in addressing the comments from the previous round of reviews. The paper is much improved. I have no further concerns.

Author Response

Thank you for the positive feedback.

Reviewer 3 Report

The part of cybersickness was improved as I suggested. 

I do not understand why PRISMA criteria was not considered. This condition is mandatory in review for MDPI, as guidelines suggest. The section 1. Prevision work, already used criteria similar to Prisma, you do not need to re-write the criteria, but only add a table to summarize it and the articles selection.

Thanks for collaboration

Author Response

Thank you for the positive feedback on cybersickness. Regarding PRISMA, we acknowledge that we use similar criteria but we did not use the precise PRISMA methodology so writing that would be misleading at this point of the process.